# Sepsis recording in primary care electronic health records, linked hospital episodes and mortality records: Population-based cohort study in England

**Emma Rezel-Potts**[1,2]*, **Martin C. Gulliford**[1,2], **the Safe AB Study Group**[¶]

**1** School of Population Health and Environmental Sciences, Guy's Campus, King's College London, London, United Kingdom, **2** National Institute for Health Research Biomedical Research Centre at Guy's and St Thomas' Hospitals London, Great Maze Pond, London, United Kingdom

¶ Membership of the Safe AB Study Group is provided in the Acknowledgements.
* emma.rezel-potts@kcl.ac.uk

**Data Availability Statement:** The study is based on data from the Clinical Practice Research Datalink (CPRD) obtained under license from the

## Abstract

### Background

Sepsis is a growing concern for health systems, but the epidemiology of sepsis is poorly characterised. We evaluated sepsis recording across primary care electronic records, hospital episodes and mortality registrations.

### Methods and findings

Cohort study including 378 general practices in England from Clinical Practice Research Datalink (CPRD) GOLD database from 2002–2017 with 36,209,676 patient-years of follow-up with linked Hospital Episode Statistics (HES) and Office for National Statistics (ONS) mortality registrations. Incident sepsis episodes were identified for each source. Concurrent records from different sources were identified and age-standardised and age-specific incidence rates compared. Logistic regression analysis evaluated associations of gender, age-group, fifth of deprivation and period of diagnosis with concurrent sepsis recording.

There were 20,206 first episodes of sepsis from primary care, 20,278 from HES and 13,972 from ONS. There were 4,117 (20%) first HES sepsis events and 2,438 (17%) mortality records concurrent with incident primary care sepsis records within 30 days. Concurrent HES and primary care records of sepsis within 30 days before or after first diagnosis were higher at younger or older ages and for patients with the most recent period of diagnosis. Those diagnosed during 2007:2011 were less likely to have a concurrent HES record given CPRD compared to those diagnosed during 2012–2017 (odd ratio 0.65, 95% confidence interval 0.60–0.70). At age 85 and older, primary care incidence was 5.22 per 1,000 patient years (95% CI 1.75–11.97) in men and 3.55 (0.87–9.58) in women which increased to 10.09 (4.86–18.51) for men and 7.22 (2.96–14.72) for women after inclusion of all three sources.

UK Medicines and Healthcare Products Regulatory Agency (MHRA); however, the interpretation and conclusions contained in this report are those of the authors alone. Requests for access to data from the study should be addressed to the corresponding author at emma.rezel-potts@kcl.ac.uk. All proposals requesting data access will require approval from CPRD before data release.

**Funding:** The study is funded by the National Institute for Health Research (NIHR) Health Services and Delivery Programme (16/116/46). MG was supported by the NIHR Biomedical Research Centre at Guy's and St Thomas' Hospitals. The views expressed are those of the authors and not necessarily those of the NHS, the NIHR, or the Department of Health. The funder of the study had no role in study design, data collection, data analysis, data interpretation, or writing of the report. The authors had full access to all the data in the study and both authors shared final responsibility for the decision to submit for publication.

**Competing interests:** The authors have no conflicts of interest.

## Conclusion

Explicit recording of 'sepsis' is inconsistent across healthcare sectors with a high proportion of non-concurrent records. Incidence estimates are higher when linked data are analysed.

## Introduction

Sepsis is a growing concern for health systems. In the UK, sepsis is estimated to account for 36,900 deaths and 123,000 hospital admissions annually [1]. The Global Burden of Disease Study estimated that there were nearly 50 million incident cases of sepsis worldwide in 2017, with 11 million deaths representing 19·7% of global deaths [2]. The term sepsis was introduced by ancient Greek physicians, but only in recent years has sepsis come to be defined as a syndrome resulting from the interaction between an acute infection and host response leading to new organ dysfunction [3]. Sepsis is an intermediate state that links an infection, or an infection-causing condition, to adverse health outcomes. The term sepsis is now more commonly used than the term 'septicaemia', which refers to blood-stream infection. In the health care systems of high-income countries, records of 'sepsis' have been increasing in both hospital and primary care settings [4–6]. A study from the U.S. Massachusetts General Hospital [7] found that recording of severe sepsis or septic shock increased by 706% in the decade between 2003 and 2012, while objective markers of severe infection, including positive blood cultures, remained stable or decreased. In a large study from UK primary care the incidence of sepsis diagnoses increased throughout the period from 2002 to 2017, with an especially rapid increase since 2011 [6]. Alongside increasing use of the term sepsis, case definitions have expanded to include patients with evidence of both acute infection and acute organ dysfunction as having 'implicit sepsis' even when sepsis was not explicitly diagnosed [2,8].

The NHS in England produced a cross-sectoral plan to improve outcomes for patients with sepsis [9]. This aimed to improve training between different healthcare professional groups and reduce differences in coding practices between organisations. A National Early Warning Score (NEWS) (updated in 2017 to NEWS2) was also rolled out to improve early recognition of signs of sepsis but this has been accompanied by concerns for false positive alerts and possible over-diagnosis of sepsis [10].

Accurate surveillance of sepsis and other bacterial infections is of importance in the context of antimicrobial resistance which can limit options for effective treatment [11]. Antimicrobial stewardship is a strategy to optimise antibiotic prescribing practices to prevent increasing antimicrobial resistance, aiming to strike a balance between effective management of suspected infection and avoidance of inappropriate or unnecessary antibiotic use [12]. The recent increase in sepsis has been accompanied by heightened awareness of antimicrobial resistance and antimicrobial stewardship, raising safety concerns about the potential for increased rates of serious bacterial infections if antibiotics are not used when needed [13].

Electronic health records (EHRs) provide an important data resource for epidemiological research and health surveillance, especially in health care settings. Data linkage provides opportunities to enhance the completeness of ascertainment of health events across health service sectors and population health registries. The advantages of linked records for case ascertainment have been demonstrated for long-term conditions [14–17], but research into the use of linked records for the evaluation of infectious diseases has been limited. However, studies investigating the incidence of community-acquired pneumonia indicate that primary care data alone may lead to under-estimation of the burden of infections [18,19].

This study aimed to exploit data linkage to evaluate the recording of sepsis across primary care EHRs, hospital episodes and mortality registrations for individual patients registered at general practices in England. We conducted a population-based cohort study to compare simultaneous recording of sepsis in primary care, hospital episodes and mortality data and to estimate the incidence of sepsis from different data sources.

## Methods

### Study population & data sources

The study employed the UK Clinical Practice Research Datalink (CPRD) GOLD database. The CPRD GOLD is a primary care database of anonymised electronic health records for general practices in the UK. The high quality of CPRD GOLD data is well-established [20]. CPRD GOLD has a coverage of some 11.3 million patients, including approximately 7% of UK population, of which it is broadly representative in terms of age and sex [21]. Consenting practices in England participate in a data linkage scheme [22]. Approximately 74% of all CPRD GOLD practices in England are eligible for linkage. Linkages are available for the Hospital Episode Statistics (HES) and mortality registration data from the Office for National Statistics (ONS). HES admitted patient care data include admission and discharge dates and diagnostic data coded using the International Classification of Diseases 10th revision (ICD10). Mortality registration data include information for the date and causes of death coded using ICD10. ONS identifies one underlying cause of death and secondary causes of death including up to 15 additional causes of death. Linked area-based measures of deprivation include the Indices of Multiple Deprivation (IMD) based on a weighted profile of indicators [23]. We employed deprivation for the general practice postcode for this study because of the low proportion of missing values. The protocol was approved by the CPRD Independent Scientific Advisory Committee (ISAC protocol 18-041R).

### Main measures

We employed data from the January 2019 release of CPRD GOLD using the 'Set 16' linkage dataset. This release of CPRD GOLD included data for 16.07 million patients of whom 8.89 million were eligible for linkage in Set 16. We included patient records between 1st January 2002 to 31st December 2017. In 2002, there were 4.48 million patients in CPRD GOLD of whom 2.48 million were eligible for HES linkage; in 2017, there were 3.53 million patients in CPRD GOLD with 0.93 million eligible for HES linkage. The start of the patient record was the later of the patient registration date or the date the general practice joined CPRD. The end of the patient record was the earliest of the last data collection date, the end of registration and the date of death. We evaluated first records of sepsis more than 12 months after start of registration in primary care electronic health records, or as a primary diagnosis in HES, or sepsis as any mentioned cause of death in mortality records. In UK primary care records, diagnoses recorded at consultations or referrals to or from hospitals were coded, at the time of this study, using Read codes. We identified sepsis records using a list of 77 eligible Read codes. Incident episodes of sepsis in CPRD were recorded using 55 Read codes with four codes accounting for 92% of events, including 'Sepsis' (two codes) (64%), 'Septicaemia' (18%), and 'Urosepsis' (10%). In HES and death registry records, sepsis diagnoses and sepsis deaths were defined using 23 ICD10 codes for sepsis. In HES records we evaluated the primary diagnosis, which accounts for the majority of the length of stay of the episode, with other diagnoses being referred to as comorbidities [24]. Incident diagnoses of sepsis in HES were coded with 20 ICD10 codes with three codes accounting for 89% of events, including 'Sepsis, unspecified' (72%), 'Sepsis due to other Gram-negative organisms' (13%) and 'Sepsis due to Staphylococcus

aureus' (5%). In mortality data, we included all mentioned causes of death because sepsis may be part of a sequence of morbid events and not always an underlying cause of death [25]. 'Sepsis, unspecified' accounted for 93% of causes of death among those in the ONS death registry with sepsis as any mentioned cause of death.

## Analysis

Incident sepsis events were identified for each data source. We calculated person-time at risk from the start to the end of the patient record. Person-time was grouped by gender and age-group from zero to four, five to nine and 10 to 14 and then 10 years age groups up to 85 years and over. Incidence and mortality rates were age-standardised using the European standard population for reference. We searched for concurrent events across data sources using a 30-day time-window. We calculated age-specific incidence rates using primary care EHRs and then adding HES records, mortality records or both. Finally, we fitted a logistic regression model to evaluate associations of gender, age-group, fifth of deprivation and period of diagnosis with concurrent sepsis recording. All data were analysed in R, version 3.6.3.

## Sensitivity analyses

In order to consider recurrent sepsis events, we conducted a sensitivity analysis using CPRD and HES where incident events were first sepsis records during each calendar year during the study period. We also evaluated the effect of extending the time-window for concurrent events to 90 and 150 days and the effect of including first and subsequent events.

## Results

### Incidence of sepsis in each data source

There were 4,081,214 registered patients at 378 general practices from 2002 to 2017, who were eligible for linkage and contributed 36,209,676 patient-years (PY) of follow-up. There were 20,206 patients with a first episode of sepsis recorded in primary care EHRs, 20,278 in HES and 13,972 in ONS. The characteristics of patients with sepsis from each data source are presented in Table 1. Each data source showed a slightly higher proportion of females than males (CPRD: 51%; HES: 52%; ONS: 55%). The frequency of sepsis increased with age, with a maximum over 75 years of age. The most deprived IMD quintile had the highest proportion of sepsis cases for each data source for men and women (25% to 26%) and the least deprived quintile consistently had the lowest (15% to 17%). The number of first sepsis events increased over time for primary care and hospital records with more first episode sepsis cases in the period 2012:2017 for both males and females. In CPRD, the period 2012:2017 accounted for 44% of sepsis episodes for men and 42% for women; in HES the latest period accounted for 50% of cases for both genders. This increasing trend was not apparent in mortality records. The period with the highest number of sepsis deaths in the ONS file was 2007:2011.

Annual age- and gender-standardised incidence of sepsis is shown for each data source in Fig 1. Primary care electronic records showed a steady increase in sepsis cases over the study period, with an acceleration from 2012 to 2017. In the CPRD in 2002, the age-standardised incidence was 0.35 (95% confidence interval 0.32 to 0.39) per 1,000 PYs in males and 0.34 (0.30 to 0.37) per 1,000 PYs in females. By 2017, this increased to 1.15 (1.04 to 1.26) per 1,000 PYs among males and 1.10 per 1,000 PYs among females (95% CI 1.00 to 1.19). Consistent with primary care data, HES data showed a steep increase in sepsis cases over the study period, particularly from 2012 to 2017. In HES records for 2002, the age-standardised incidence was 0.29 (0.25 to 0.32) per 1,000 PYs among males and 0.25 (0.22 to 0.28) per 1,000 PYs among

**Table 1. Characteristics of patients with incident sepsis events in three data sources.** Figures are frequencies (% of column total).

| | | Primary care (CPRD) | | Hospital episodes (HES) | | Mortality data (ONS) | |
|---|---|---|---|---|---|---|---|
| | | Male | Female | Male | Female | Male | Female |
| **Total** | | 9 893 (49) | 10 313 (51) | 9 796 (48) | 10 482 (52) | 6 245 (45) | 7 727 (55) |
| **Age-group (years)** | 0–4 | 137 (1) | 138 (1) | 122 (1) | 142 (1) | 5 (0) | 5 (0) |
| | 5–14 | 194 (2) | 181 (2 | 131 (1) | 82 (1) | 13 (0) | 6 (0) |
| | 15–24 | 230 (2) | 293 (3) | 114 (1) | 121 (2) | 19 (0) | 17 (0) |
| | 25–34 | 257 (3) | 464 (4) | 191 (2) | 238 (2) | 33 (0) | 19 (0) |
| | 35–44 | 461 (5) | 667 (6) | 353 (4) | 430 (4) | 91 (1) | 83 (1) |
| | 45–54 | 786 (8) | 911 (9) | 651 (7) | 840 (8) | 219 (4) | 219 (3) |
| | 55–64 | 1 410 (14) | 1 351 (13) | 1 298 (13) | 1 180 (11) | 520 (8) | 468 (6) |
| | 65–74 | 2 186 (22) | 1 756 (17) | 2 178 (22) | 1 809 (17) | 1 172 19) | 962 (12) |
| | 75–84 | 2 639 (27) | 2 281 (22) | 2 855 (29) | 2 661 (25) | 2 169 (35) | 2 423 (31) |
| | 85+ | 1 593 (16) | 2 271 (22) | 1 903 (19) | 2 979 (28) | 2 004 (32) | 3 525 (46) |
| **Deprivation fifth** | Least deprived | 1 570 (16) | 1 502 (15) | 1 630 (17) | 1 662 (16) | 923 (15) | 1 123 (15) |
| | 2nd | 1 570 (16) | 1 502 (15) | 1 630 (17) | 1 662 (16) | 923 (15) | 1 123 (15) |
| | 3rd | 2 126 (21) | 2 403 (23) | 1 892 (19) | 2 004 (19) | 1 258 (20) | 1 525 (20) |
| | 4th | 1 852 (19) | 1 817 (18) | 1 892 (19) | 1 960 (19) | 1 150 (18) | 1 408 (18) |
| | Most deprived | 1 889 (19) | 1 914 (19) | 1 944 (20) | 2 178 (21) | 1 308 (21) | 1 657 (21) |
| | Data missing | 1 (0) | 2 (0) | 1 (0) | 4 (0) | 6 (0) | 4 (0) |
| **Period** | 2002:2006 | 2 271 (23) | 2 604 (25) | 1 911 (19) | 2 133 (20) | 2 077 (33) | 2 550 (33) |
| | 2007:2011 | 3 234 (33) | 3 422 (33) | 2 964 (30) | 3 155 (30) | 2 398 (38) | 3 086 (40) |
| | 2012:2017 | 4 388 (44) | 4 287 (42) | 4 921 (50) | 5 194 (50) | 1 770 (28) | 2 091 (27) |

CPRD, Clinical Practice Research Datalink; HES, Hospital Episode Statistics; ONS, Office for National Statistics.

females. By 2017, this had increased to 2.52 (2.35 to 2.68) per 1,000 PYs among males and 2.10 (1.96 to 2.23) per 1,000 PYs among females. In contrast to primary care and hospital records, there was no consistent trend in sepsis recording as a cause of death during the study period. In 2002 sepsis mortality was 0.34 (0.31 to 0.38) per 1,000 PYs among males and 0.28 (0.26 to 0.31) per 1,000 PYs among females. By 2017, this had risen to 0.42 (0.35 to 0.49) per 1,000 PYs among males and 0.30 (0.25 to 0.35) per 1,000 PYs among females although the highest peak in mortality rate was in 2006 for females and 2007 for males. S1 Fig shows equivalent age-standardised rates when first sepsis records in each calendar year are included, rather than only the first in the study period.

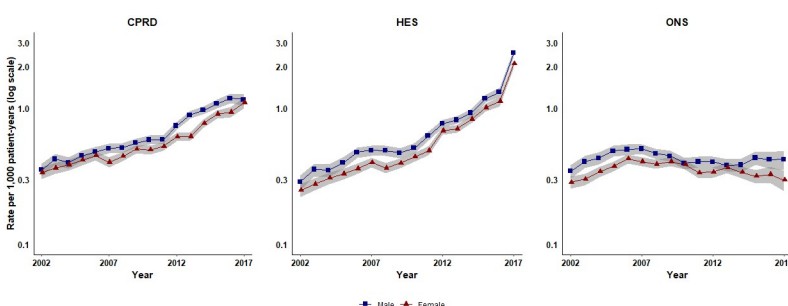

**Fig 1. Age-standardised incidence of first cases of sepsis in CPRD and HES and age-standardised sepsis mortality in ONS reported from 2002–2017, stratified by gender.** CPRD, Clinical Practice Research Datalink; HES, Hospital Episode Statistics; ONS, Office for National Statistics.

## Concurrent recording of sepsis events across data sources

There were 4,117 (20%) first HES sepsis events and 2,438 (17%) mortality records that were concurrent with incident primary care sepsis records, based on a 30-day time-window (Fig 2 and S1 Table). Among the 13,972 deceased patients with sepsis listed as any cause of death, 2,438 (17%) had an incident sepsis event recorded in primary care EHRs and 3,397 (24%) had incident sepsis events recorded in HES in the same period. Only 614 patients had index sepsis events recorded across all three data sources within 30 days of the date of event or date of death. We evaluated whether extending the time-window influenced conclusions. Including both first and subsequent sepsis records and a 90-day time window gave 4,770 (24%) HES records and 2,635 (19%) of mortality records concurrent with an incident primary care sepsis record. (S1 Table and S3 and S4 Figs). Extending this to a 150-day time window indicated similar levels of concurrence (S1 Table and S5 Fig).

Logistic regression analysis of variables associated with concurrent recording in more than one data source for patients with sepsis found that among patients with sepsis events recorded in primary care, concurrent HES records of sepsis within 30 days before or after first diagnosis was higher at younger or older ages and for patients with the most recent period of diagnosis (Fig 3). For index sepsis events recorded in HES, concurrent recording in primary care EHRs was not consistently associated with age, deprivation nor period. Among patients with sepsis events recorded in ONS, there were lower odds of a concurrent CPRD record of sepsis for the age range 35 to 54 years, being registered with a practice in the least deprived quintile and the most recent period of diagnosis (Fig 4).

## 3.4 Incidence of sepsis from linked data sources

Table 2 shows age-specific sepsis incidence rate per 1,000 patient-years from primary care EHRs and for primary care EHRs combined with HES and ONS, stratified. With primary care EHRs alone, sepsis incidence rates increased from 0.19 in males and 0.20 in females per 1,000 PY under five years of age, to 5.22 in males and 3.55 in females at 85 years and older. Estimated incidence rates were substantially higher when either HES or mortality records were included. When all three data sources were combined the incidence of sepsis was 0.32 per 1,000 PY in

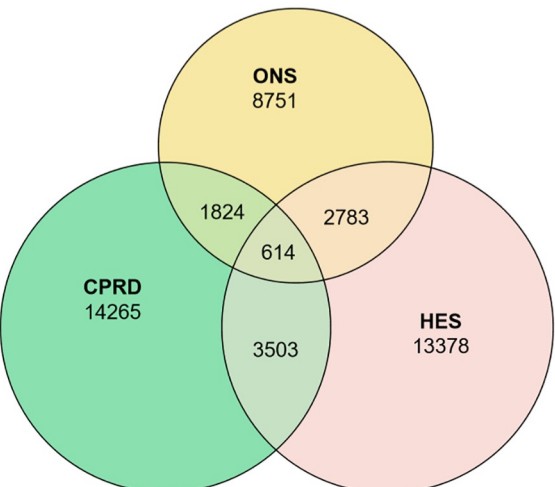

**Fig 2. Incident first sepsis events in CPRD, HES and ONS using a 30-day time-window to evaluate concurrence.**
CPRD, Clinical Practice Research Datalink; HES, Hospital Episode Statistics; ONS, Office for National Statistics.

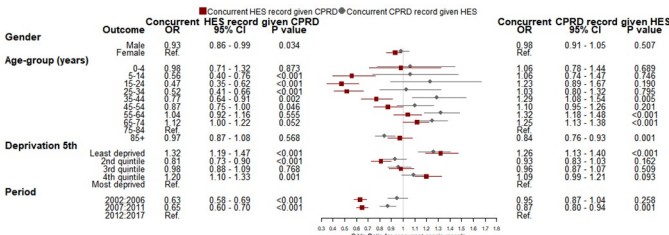

**Fig 3. Logistic regression model of variables associated with the outcome of a) concurrent index sepsis recording HES given sepsis event in CPRD and b) concurrent index sepsis recording in CPRD given index sepsis event in HES.** CPRD, Clinical Practice Research Datalink; HES, Hospital Episode Statistics; ONS, Office for National Statistics.

males and 0.36 per 1,000 PY in females under five years increasing to 10.09 per 1,000 PY in males and 7.22 per 1,000 PY in females at age 85 years and older. S2 Fig shows equivalent age-specific incidence when first sepsis records in each calendar year, rather than the first in the study period, were included.

## Discussion

### Summary of results

This study evaluated sepsis recording in primary care electronic records, hospital episodes and mortality records for a large population registered with general practices in England. We found that, over a 16-year period, a similar number of incident sepsis events were recorded in primary care and hospital records, However, a high proportion of these records were not concurrent across data sources: the majority of sepsis events recorded in primary care were not recorded in hospital episodes, and the majority of hospital episodes were not recorded in primary care. This conclusion held even when events were evaluated over a longer time-window or if recurrent as well as incident events were included. There were a smaller number of records of sepsis from mortality registration but a majority of these were not associated with concurrent primary care or hospital records for sepsis. Analyses of associations between

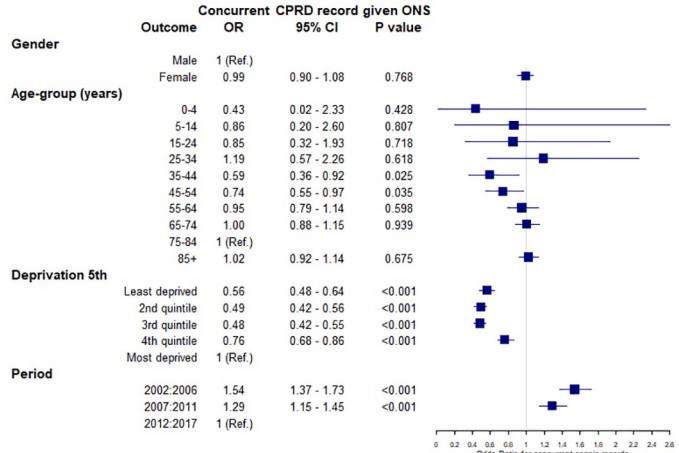

**Fig 4. Logistic regression model of variables associated with the outcome of concurrent index sepsis recording in CPRD given recording of sepsis as any cause of death in ONS.** CPRD, Clinical Practice Research Datalink; ONS, Office for National Statistics.

**Table 2. Age-specific sepsis incidence rate (exact Poisson 95% confidence intervals) per 1,000 patient-years in each CPRD and CPRD combined with HES and ONS, stratified by gender.**

| Age group (years) | CPRD | | CPRD + HES | | CPRD + ONS | | CPRD + ONS + HES | |
|---|---|---|---|---|---|---|---|---|
| | Male | Female | Male | Female | Male | Female | Male | Female |
| 0 to 4 | 0.19 (0.00 to 4.08) | 0.20 (0.00 to 4.10) | 0.32 (0.00 to 4.33) | 0.36 (0.00 to 4.40) | 0.20 (0.00 to 4.09) | 0.21 (0.00 to 4.11) | 0.32 (0.00 to 4.33) | 0.36 (0.00 to 4.40) |
| 5 to14 | 0.09 (0.00 to 3.87) | 0.09 (0.00 to 3.87) | 0.13 (0.00 to 3.96) | 0.12 (0.00 to 3.93) | 0.09 (0.00 to 3.88) | 0.09 (0.00 to 3.88) | 0.13 (0.00 to 3.96) | 0.12 (0.00 to 3.93) |
| 15 to 24 | 0.11 (0.00 to 3.92) | 0.16 (0.00 to 4.02) | 0.14 (0.00 to 3.99) | 0.21 (0.00 to 4.11) | 0.12 (0.00 to 3.94) | 0.17 (0.00 to 4.03) | 0.14 (0.00 to 3.99) | 0.21 (0.00 to 4.11) |
| 25 to 34 | 0.12 (0.00 to 3.93) | 0.21 (0.00 to 4.12) | 0.18 (0.00 to 4.05) | 0.29 (0.00 to 4.28) | 0.13 (0.00 to 3.95) | 0.22 (0.00 to 4.13) | 0.18 (0.00 to 4.05) | 0.29 (0.00 to 4.28) |
| 35 to 44 | 0.17 (0.00 to 4.03) | 0.25 (0.00 to 4.19) | 0.26 (0.00 to 4.21) | 0.36 (0.00 to 4.41) | 0.19 (0.00 to 4.08) | 0.27 (0.00 to 4.24) | 0.26 (0.00 to 4.21) | 0.36 (0.00 to 4.41) |
| 45 to 54 | 0.29 (0.00 to 4.27) | 0.35 (0.00 to 4.38) | 0.46 (0.00 to 4.60) | 0.57 (0.00 to 4.81) | 0.35 (0.00 to 4.39) | 0.42 (0.00 to 4.52) | 0.46 (0.00 to 4.60) | 0.57 (0.00 to 4.81) |
| 55 to 64 | 0.61 (0.00 to 4.87) | 0.59 (0.00 to 4.84) | 0.99 (0.02 to 5.55) | 0.93 (0.02 to 5.45) | 0.78 (0.01 to 5.19) | 0.74 (0.01 to 5.11) | 0.99 (0.02 to 5.55) | 0.93 (0.02 to 5.45) |
| 65 to 74 | 1.30 (0.07 to 6.09) | 0.98 (0.02 to 5.53) | 2.19 (0.31 to 7.53) | 1.68 (0.15 to 6.72) | 1.83 (0.19 to 6.95) | 1.39 (0.08 to 6.24) | 2.19 (0.31 to 7.53) | 1.68 (0.15 to 6.72) |
| 75 to 84 | 2.65 (0.47 to 8.23) | 1.73 (0.16 to 6.79) | 4.74 (1.86 to 11.29) | 3.23 (0.72 to 9.12) | 4.35 (1.27 to 10.75) | 3.17 (0.69 to 9.02) | 4.74 (1.48 to 11.29) | 3.23 (0.72 to 9.12) |
| 85+ | 5.22 (1.75 to 11.97) | 3.55 (0.87 to 9.58) | 10.09 (4.86 to 18.51) | 7.22 (2.96 to 14.72) | 10.42 (5.09 to 18.93) | 7.93 (3.41 to 15.67) | 10.09 (4.86 to 18.51) | 7.22 (2.96 to 14.72) |

CPRD, Clinical Practice Research Datalink; HES, Hospital Episode Statistics; ONS, Office for National Statistics.

concurrent recording between sources and available patient covariates were not consistent across linkages, suggesting that coding variations were largely unexplained. Estimates for age-specific incidence rates may be up to twice as high if linked data sources are employed.

These observations extend our previous research into the probability (incidence) of sepsis after infection consultations in primary care [26]. The occurrence of sepsis is highly age-dependent, with the probability of sepsis after infection consultations in primary care being greatest at 85 years or older. In this age-group, the probability of sepsis was estimated to be 0.003 if antibiotics were not prescribed and 0.0005 if antibiotics were prescribed (Ref Fig 2). Estimates were generally slightly greater when linked data were employed (Ref S4 Fig) but events recorded from HES or mortality statistics were less frequently associated with general practice consultations than those recorded into CPRD data (Ref S3 Fig). This confirms that the incidence of sepsis will be higher when data linkage is employed but events recorded into different data sources may show distinct characteristics.

## Interpretation

Current estimates suggest that about 50–70% of all sepsis events are community-acquired [27,28]. Clinical guidelines recommend that patients with suspected sepsis should be referred for management in hospital, which suggests that most patients seen in primary care may be admitted to hospital [1]. However, some patients with sepsis may access hospital services directly without first presenting in primary care. Investigations in hospital may identify underlying causes for sepsis, which might be coded as the reason for admission. Just under half of severe sepsis cases admitted to intensive care in England and Wales are associated with a fatal outcome [29]. It may be expected that a sepsis diagnosis will be communicated to the patient's general practice or, in the event of a fatal outcome, recorded into mortality records. Our results

indicate that the process of recording sepsis episodes across different health information systems is highly inconsistent. Health professionals may make varying use of the concept of sepsis in the clinical recording of patients' conditions. 'Sepsis' may sometimes form an element of the clinical narrative but, on other occasions, an underlying cause may be given greater prominence, as in the COVID-19 pandemic. Increased sepsis recording in CPRD and HES in the period 2012:2017 corresponds with strategies implemented by the NHS to improve the standardisation of management of the deterioration in sepsis, such as NEWS which began its first iteration in 2012. [10]. Conversely, our findings show that ONS death registry records were not in parallel with the CPRD and HES records, indicating a drop in sepsis deaths during this time. It may be that NHS strategies have improved the severity of sepsis outcomes or that current processes of certification and registration of deaths are failing to record sepsis as an underlying or secondary cause of death.

## Limitations

The present results derive from a large, representative population. However, a disadvantage of using linked, rather than stand-alone CPRD records is that linkage eligibility restrictions reduces the sample size and possibly representativeness. GPs must be eligible for linkage which requires meeting standards of data completeness, thus biasing the practices within the linked sample toward those that record disease events with greater accuracy. It is also possible that there are some inaccuracies with linkage across sources. Discrepancies between CPRD GOLD and the ONS death data have been highlighted, particularly in the years prior to 2013 [30]. This study does not differentiate between community-acquired and hospital-acquired sepsis. Further research is required to understand how sepsis incidence in the combined sources can reveal more about the impact of antimicrobial stewardship strategies.

## Findings in relation to other studies

This study is broadly consistent with the growing body of literature that advocates the use of linked data sources [15–18]. However, it also indicates that stand-alone CPRD data may provide accurate estimates of changes in the burden of sepsis. Millet et al. (2016) found that population-averaged community-acquired pneumonia incidence was 39% higher using linked rather than stand-alone data [18]. It may be that increased awareness and standardisation of detection and recording have prevented such discrepancies being observed for sepsis.

## Conclusion

Analysing linked data enhances the completeness of ascertainment of health events across health service sectors and population health registries particularly for age-groups at highest risk. Further standardisation of coding practices across linked sources in addition to more timely and accurate recording of secondary care and mortality events in GP records would help to improve comparability. Further research is required to investigate the reasons for any divergent trends across the data sources and to differentiate trends in community-acquired versus hospital-acquired sepsis for more effective monitoring of antibiotic stewardship strategies.

## Supporting information

**S1 Checklist. Checklist of items that should be included in reports of *cohort studies*.**
(DOC)

**S1 Fig. Age-standardised rates of sepsis events including first events in each calendar-year.**
(TIF)

**S2 Fig. Age-specified rates of sepsis events including first events in each calendar-year.**
(TIF)

**S3 Fig. Sepsis cases in CPRD, HES, ONS (concurrent cases have index or recurrent HES sepsis events in 30 days before or after index event in CPRD or index or recurrent sepsis events within 30 days before date of death in ONS death registry).**
(TIF)

**S4 Fig. Incident first sepsis events in CPRD, HES and ONS using a 90-day time-window to evaluate concurrence.**
(TIF)

**S5 Fig. Incident first sepsis events in CPRD, HES and ONS using a 150-day time-window to evaluate concurrence.**
(TIF)

**S1 Table. Concurrence of sepsis events in CPRD, ONS, HES using 30-day, 90- day and 150-day time-windows and either first or first and subsequent events.** Figures are frequencies (% of column total).
(DOCX)

## Acknowledgments

### Declarations

The SafeAB Study Group includes Martin C. Gulliford, Caroline Burgess, Vasa Curcin, James Shearer, Judith Charlton, Joanne R. Winter, Xiaohui Sun, Emma Rezel-Potts, Catey Bunce, Robin Fox, Paul Little, Alastair D Hay, Michael V. Moore and Mark Ashworth. The lead author for the SafeAB Study Group is Martin C. Gulliford (martin.gulliford@kcl.ac.uk).

## Author Contributions

**Conceptualization:** Martin C. Gulliford.

**Data curation:** Emma Rezel-Potts.

**Formal analysis:** Emma Rezel-Potts.

**Funding acquisition:** Martin C. Gulliford.

**Methodology:** Emma Rezel-Potts.

**Software:** Emma Rezel-Potts.

**Supervision:** Martin C. Gulliford.

**Validation:** Martin C. Gulliford.

**Visualization:** Emma Rezel-Potts.

**Writing – original draft:** Emma Rezel-Potts.

**Writing – review & editing:** Martin C. Gulliford.

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
