## [Decision Letter · Decision Letter 0]

12 Nov 2020

PONE-D-20-21285

Sepsis Recording in Primary Care Electronic Health Records, Linked Hospital Episodes and Mortality Records: Population-based Cohort Study in England

PLOS ONE

Dear Dr. Rezel-Potts,

Thank you for submitting your manuscript to PLOS ONE. After careful consideration, we feel that it has merit but does not fully meet PLOS ONE’s publication criteria as it currently stands. Therefore, we invite you to submit a revised version of the manuscript that addresses the points raised during the review process.

We look forward to receiving your revised manuscript.

Kind regards,

Sreeram V. Ramagopalan

Academic Editor

PLOS ONE

Journal Requirements:

2. As part of your revision, please complete and submit a copy of the STROBE Guidelines checklist, a document that aims to improve reporting of cohort studies for purposes of post-publication data analysis and reproducibility. Please include your completed checklist as a Supporting Information file. Note that if your paper is accepted for publication, this checklist will be published as part of your article.

4. One of the noted authors is a group or consortium [SafeAB Study Group]. In addition to naming the author group, please list the individual authors and affiliations within this group in the acknowledgments section of your manuscript. Please also indicate clearly a lead author for this group along with a contact email address.

Reviewers' comments:

Reviewer's Responses to Questions

**Comments to the Author**

1. Is the manuscript technically sound, and do the data support the conclusions?

Reviewer #1: Yes

2. Has the statistical analysis been performed appropriately and rigorously? 

Reviewer #1: Yes

3. Have the authors made all data underlying the findings in their manuscript fully available?

Reviewer #1: No

4. Is the manuscript presented in an intelligible fashion and written in standard English?

Reviewer #1: Yes

5. Review Comments to the Author

Reviewer #1: Is the manuscript technically sound, and do the data support the conclusions?

Yes, on the whole, the paper investigates concurrent recording of sepsis in CPRD Gold, and linked ONS and HES data.

I am concerned that the method be missing some concurrent records. This may be more to do with description, rather than actual method.

In the situation where a person has attended hospital twice for sepsis, 100 days apart both times recorded in HES. Only the second time does the GP enter a structured record of the sepsis. Neither time does the patient die. Would the paper record no agreement between the HES and CPRD, even though the second HES visit was recorded in the GP record?

The sensitivity analysis would also not pick up such a situation (assuming the calendar year did not change).

Could delays between initial diagnosis for sepsis and death with sepsis as a cause be longer than 30/90 days?

Has the statistical analysis been performed appropriately and rigorously?

The study is relevant for people completing studies in CPRD that are considering linkage to identify sepsis events. I would expect that in most studies, linkage would be used to add events whenever they occur, regardless of whether there is a concurrent event in the CPRD Gold record.

Such researchers therefore need to understand how linkage would impact their sample size. A flow chart showing how many pts in CPRD, how many eligible for linkage, and how this changes over time could benefit the paper.

Comparisons of sepsis incidence between the unlinked CPRD Gold data and the linked subset would add some useful comparisons. What would the estimate of sepsis incidence be if only unlinked CPRD Gold data was used. Perhaps reporting sepsis incidence by year for each dataset, ignoring whether sepsis recording was concurrent, would be useful for researchers. Similar to figure 2 in https://onlinelibrary.wiley.com/doi/full/10.1002/pds.4747

The conclusion that the odds of recording sepsis in CPRD given a record in the ONS is higher in 2002-2006 and 2007-2011 is an interesting finding. What could be driving that? Why are there lower numbers of ONS sepsis death records 2012-2017?

Have the authors made all data underlying the findings in their manuscript fully available?

No, this is not possible with CPRD data.

Is the manuscript presented in an intelligible fashion and written in standard English?

Yes.

This sentence in the abstract is hard to understand, perhaps better broken down.

Concurrent HES and primary care records of sepsis within 30 days before or after first

diagnosis were higher at younger or older ages and for patients with the most recent

period of diagnosis with those diagnosed during 2007:2011 less likely to have a

concurrent HES record given CPRD compared to those diagnosed during 2012-2017

(odd ratio 0.65, 95% confidence interval 0.60 – 0.70).

Additional comments

In the introduction it would be useful to note the many NHS initiatives that have occurred over time to improve sepsis management and recording, such as the sepsis action plan.

The authors mentioned community acquired sepsis a number of times. Is this assuming that primary care would be recognizing community acquired sepsis better than hospitals? It seems likely that community acquired sepsis would also end up directly in hospital (as the authors note). And CPRD Gold should be recording sepsis diagnoses from hospitals. So location of diagnosis record does not seem a good measure of whether it is community or hospital acquired.

Figure 2: the relative sizes of the venn diagram does not represent very well the small number of sepsis co-occurrence in the ONS data. Perhaps a chart that graphically represents size could be better here.

6. PLOS authors have the option to publish the peer review history of their article (what does this mean?). If published, this will include your full peer review and any attached files.

Reviewer #1: **Yes: **S Wilkinson

---

## [Author Response · Author response to Decision Letter 0]

3 Dec 2020

Sepsis Recording in Primary Care Electronic Health Records, Linked Hospital Episodes and Mortality Records: Population-based Cohort Study in England (PONE-D-20-21285)

Requests from Editors: 

Thank you, we have followed PLOS ONE’s style requirements and renamed files according to guidelines. 

2. As part of your revision, please complete and submit a copy of the STROBE Guidelines checklist, a document that aims to improve reporting of cohort studies for purposes of post-publication data analysis and reproducibility. Please include your completed checklist as a Supporting Information file. Note that if your paper is accepted for publication, this checklist will be published as part of your article.

Thank you, we have submitted a copy of the STROBE Guidelines checklist as requested.

There are legal restrictions on sharing this data publicly, please see response to 3a and further details in our revised cover letter. Also, the reviewer response to question, “Have the authors made all data underlying the findings in their manuscript fully available?” which was “No, this is not possible with CPRD data”.

The study is based on data from the Clinical Practice Research Datalink (CPRD) obtained under license from the UK Medicines and Healthcare Products Regulatory Agency (MHRA). All proposals requesting data access require approval from CPRD before data release. Data access is governed by licence as outlined here: https://cprd.com/primary-care. The purpose of the licence is to protect patient confidentiality and ensure the integrity and security of the database. CPRD is normally agreeable to releasing data in response to requests, but this is subject to ethical and scientific review, as is required for all CPRD studies. We have adjusted our Data Sharing statement in the manuscript to improve clarity on this point (pages 19 to 20, lines 422 to 427). 

See above.

4. One of the noted authors is a group or consortium [SafeAB Study Group]. In addition to naming the author group, please list the individual authors and affiliations within this group in the acknowledgments section of your manuscript. Please also indicate clearly a lead author for this group along with a contact email address.

Thank you, we have included this information in the acknowledgements section as requested (page 19, lines 396 to 401).

We have now included captions for supporting information files at the end of the manuscript (page 17, lines 376 to 390) and have updated in-text citations to match accordingly and in accordance with the guidelines.

Comments from Reviewers:

1. I am concerned that the method be missing some concurrent records. This may be more to do with description, rather than actual method. 

In the situation where a person has attended hospital twice for sepsis, 100 days apart both times recorded in HES. Only the second time does the GP enter a structured record of the sepsis. Neither time does the patient die. Would the paper record no agreement between the HES and CPRD, even though the second HES visit was recorded in the GP record? The sensitivity analysis would also not pick up such a situation (assuming the calendar year did not change). Could delays between initial diagnosis for sepsis and death with sepsis as a cause be longer than 30/90 days?

Thank you for your query. We agree that events beyond the 30 and 90 day cut-off would not be considered concurrent according to our definitions. We conducted an additional sensitivity analysis to extend the period to 150 days in accordance with the upper limit of the later phase of sepsis used by Otto et al. (2011). Results for this have now been added to S1 Table, with index concurrent events presented as an additional Venn diagram (S5 Fig). In the main report this is outlined in the “sensitivity analysis” section (page 7, line 180 to 181) and in the results section (page 11, lines 241 to 242). 

2. The study is relevant for people completing studies in CPRD that are considering linkage to identify sepsis events. I would expect that in most studies, linkage would be used to add events whenever they occur, regardless of whether there is a concurrent event in the CPRD Gold record. Such researchers therefore need to understand how linkage would impact their sample size. A flow chart showing how many pts in CPRD, how many eligible for linkage, and how this changes over time could benefit the paper.

Thank you, we now explain (pages 5 to 6, lines 137 to 142): ‘We employed data from the January 2019 release of CPRD GOLD using the ‘Set 16’ linkage dataset. This release of CPRD GOLD included data for a total of 16.07 million patients of whom 8.89 million were eligible for linkage in Set 16. We included patient records between 1st January 2002 to 31st December 2017. In 2002, there were 4.48 million patients in CPRD GOLD, of whom 2.48 million were eligible for HES linkage; in 2017, there were 3.53 million patients in CPRD GOLD, with 0.93 million eligible for HES linkage.’

3. Comparisons of sepsis incidence between the unlinked CPRD Gold data and the linked subset would add some useful comparisons. What would the estimate of sepsis incidence be if only unlinked CPRD Gold data was used. Perhaps reporting sepsis incidence by year for each dataset, ignoring whether sepsis recording was concurrent, would be useful for researchers. Similar to figure 2 in https://onlinelibrary.wiley.com/doi/full/10.1002/pds.4747

Thank you, we now explain (pages 14 to 15, lines 302 to 311): ‘These observations extend our previous research into the probability (incidence) of sepsis after infection consultations in primary care. The occurrence of sepsis is highly age-dependent, with the probability of sepsis after infection consultations in primary care being greatest at 85 years or older. In this age-group, the probability of sepsis was estimated to be 0.003 if antibiotics were not prescribed and 0.0005 if antibiotics were prescribed (Ref Figure 2). Estimates were generally slightly greater when linked data were employed (Ref Figure S4) but events recorded from HES or mortality statistics were less frequently associated with general practice consultations than those recorded into CPRD data (Ref Figure S3). This confirms that the incidence of sepsis will be higher when data linkage is employed but events recorded into different data sources may show distinct characteristics.’

4. The conclusion that the odds of recording sepsis in CPRD given a record in the ONS is higher in 2002-2006 and 2007-2011 is an interesting finding. What could be driving that? Why are there lower numbers of ONS sepsis death records 2012-2017?

We have amended the discussion to include some cautious interpretations of this finding (page 16, lines 338 to 342).

5. This sentence in the abstract is hard to understand, perhaps better broken down. Concurrent HES and primary care records of sepsis within 30 days before or after first diagnosis were higher at younger or older ages and for patients with the most recent period of diagnosis with those diagnosed during 2007:2011 less likely to have a concurrent HES record given CPRD compared to those diagnosed during 2012-2017 (odd ratio 0.65, 95% confidence interval 0.60 – 0.70).

Thank you. We have broken this down into two sentences as suggested (page 2, lines 46 to 50).

6. In the introduction it would be useful to note the many NHS initiatives that have occurred over time to improve sepsis management and recording, such as the sepsis action plan.

Thank you. We have now included details of the sepsis action plan and NEWS/NEW2 in the introduction (page 3, lines 83 to 88).

7. The authors mentioned community acquired sepsis a number of times. Is this assuming that primary care would be recognizing community acquired sepsis better than hospitals? It seems likely that community acquired sepsis would also end up directly in hospital (as the authors note). And CPRD Gold should be recording sepsis diagnoses from hospitals. So location of diagnosis record does not seem a good measure of whether it is community or hospital acquired.

We agree that our findings do not provide a good measure of whether the sepsis diagnosis is community rather than hospital acquired, identifying this as a limitation (page 16, lines 352 to 353). We feel this distinction is important to highlight because this research is undertaken within the context of antimicrobial stewardship, as stated in the introduction (page 4, lines 90 to 97). In the conclusion we have amended our recommendation for further research to establish the source of sepsis infection so that it explicitly links this back to this context (page 17, lines 373 to 374).

8. Figure 2: the relative sizes of the venn diagram does not represent very well the small number of sepsis co-occurrence in the ONS data. Perhaps a chart that graphically represents size could be better here.

Thank you for this suggestion. We have now amended all the venn diagrams so that circle sizes and overlaps reflect the relative frequencies in each (Fig 2., S3-S5 Figs).

---

## [Editor Report · Decision Letter 1]

16 Dec 2020

Sepsis recording in primary care electronic health records, linked hospital episodes and mortality records: population-based cohort study in England

PONE-D-20-21285R1

Dear Dr. Rezel-Potts,

We’re pleased to inform you that your manuscript has been judged scientifically suitable for publication and will be formally accepted for publication once it meets all outstanding technical requirements.

Kind regards,

Sreeram V. Ramagopalan

Academic Editor

PLOS ONE
---

## [Editor Report · Acceptance letter]

18 Dec 2020

PONE-D-20-21285R1 

Sepsis recording in primary care electronic health records, linked hospital episodes and mortality records: population-based cohort study in England 

Dear Dr. Rezel-Potts:

I'm pleased to inform you that your manuscript has been deemed suitable for publication in PLOS ONE. Congratulations! Your manuscript is now with our production department. 

Kind regards, 

on behalf of

Dr. Sreeram V. Ramagopalan 

Academic Editor

PLOS ONE